# A combined computational and experimental strategy identifies mutations conferring resistance to drugs targeting the BCR-ABL fusion protein

Jinxin Liu[1], Jianfeng Pei [2]* & Luhua Lai [2,3]*

Drug resistance is of increasing concern, especially during the treatments of infectious diseases and cancer. To accelerate the drug discovery process in combating issues of drug resistance, here we developed a computational and experimental strategy to predict drug resistance mutations. Using BCR-ABL as a case study, we successfully recaptured the clinically observed mutations that confer resistance imatinib, nilotinib, dasatinib, bosutinib, and ponatinib. We then experimentally tested the predicted mutants in vitro. We found that although all mutants showed weakened binding strength as expected, the binding constants alone were not a good indicator of drug resistance. Instead, the half-maximal inhibitory concentration ($IC_{50}$) was shown to be a good indicator of the incidence of the predicted mutations, together with change in catalytic efficacy. Our suggested strategy for predicting drug-resistance mutations includes the computational prediction and in vitro selection of mutants with increased $IC_{50}$ values beyond the drug safety window.

[1] The PTN Graduate Program, College of Life Sciences, Peking University, Beijing 100871, P. R. China. [2] Center for Quantitative Biology, AAIS, Peking University, Beijing 100871, P. R. China. [3] BNLMS, Peking-Tsinghua Center for Life Sciences at College of Chemistry and Molecular Engineering, Peking University, Beijing 100871, P. R. China. *email: jfpei@pku.edu.cn; lhlai@pku.edu.cn

D rug resistance is a global public health problem, especially during the treatment of infectious diseases and cancer[1]. According to the World Health Organization, drug resistance will become the leading cause of death for humans by 2050. Developing new drugs to combat issues with resistance is thus an urgent need[2]. Several mechanisms account for the increase in drug resistance in cancer, such as drug inactivation, drug-target alterations, drug efflux, DNA damage repair, cell death inhibition, epithelial–mesenchymal transition, and cytokine activity[3,4]. Among these mechanisms, mutations in target proteins that alter drug interactions are the main drivers of resistance[5–7]. Tumor cells appear to adapt to nearly all types of drugs, including blockbuster drugs such as Gleevec and Iressa[8]. Modifications of drug targets via conformational changes or altered drug interaction sites, as seen in mutations in BCR-ABL for Gleevec resistance and in EGFR for Iressa resistance, are frequently reported[9,10].

In clinical practice, the solution to drug resistance is to develop and use new generations of drugs[11]. However, the development of new generations of drugs takes time and resources, making drug resistance an impediment to the use of newly marketed drugs. Drug-resistance mutations can be found by genomic or single-cell sequencing[12,13]. Sequencing methods can often identify existing resistance mutations except for some subclones with a lower number of clones[14]. Drug resistance has been associated with a variety of cellular defects, including aberrant growth factor receptor and tumor suppressor functions. Therefore, efforts to predict drug resistance by analyzing the transcriptome and proteome are ongoing[15–17]. However, these methods cannot predict the type of mutation responsible for resistance. Azam et al.[18] performed an in vitro screen of randomly mutagenized BCR-ABL to obtain a comprehensive survey of the amino acid substitutions that confer resistance, and Corbin et al.[19] performed an extensive experimental mutational analysis of sites that might alter the sensitivity of the ABL kinase to imatinib, demonstrating a broad range of possibilities for clinical resistance. These experimental screenings were able to capture some of the drug-resistant variants in the laboratory. Mutations may not only confer drug resistance but also affecting the function and stability of proteins. Therefore, evolutionary changes in fitness landscapes caused by different mutations have also been studied[20].

Computational methods provide another way to address drug resistance, especially the identification of mechanisms of resistance. For protein targets without known three-dimensional (3D) structures, methods have been developed to predict resistance mutations directly from the primary sequence[21]. The availability of 3D structures of drug targets involved in diseases enables the use of molecular modeling, molecular dynamics, and protein-inhibitor docking methods to understand the mechanisms of resistance[22–25]. In 1998, Rosin et al.[26] randomly mutated the residues of the HIV-1 protease that are involved in binding peptide inhibitors. The binding free energy of the inhibitor and the HIV-1 protease was estimated using a rough measure of volume complementarity, and a few important mutation sites could be predicted[26,27]. Hou et al.[28] and Cao et al.[29] used MM/PBSA to calculate binding free energies based on molecular dynamics and correctly classify several experimentally known mutations. Using alchemical free-energy calculation to validate resistance for eight FDA-approved kinase inhibitors across 144 clinically identified point mutations, Hauser et al. classified clinically selected mutations into resistance mutations and susceptible mutations[25]. Liu et al. hypothesized that multi-site resistance mutations are synergistic and proposed a procedure that combines Bayesian statistical modeling and molecular dynamics to investigate the drug-resistance mutations of HIV-1 protease and reverse transcriptase[30], demonstrating the relevance of multi-site resistance mutations. Methods using sequence information and machine learning to predict possible resistance mutations have also been reported[31,32], which were confined by the training set used.

Protein kinases play main regulatory roles in nearly every aspect of cell pathways. BCR-ABL is a kind of protein kinase expressed by a fusion gene caused by a specific genetic abnormality on the chromosome 22 of the leukemia cancer cell. With long-term of BCR-ABL expression, eventually chronic myeloid leukemia (CML) is produced[33]. This fusion gene encodes a hybrid tyrosine kinase signaling protein that is always "on," causing the cell to divide uncontrollably. In the late 1990s, the small molecule kinase inhibitor STI-571 (imatinib, Gleevec) was developed by Novartis. Although it does not eradicate CML cells, it greatly limits the growth of clonal tumors. In 2000, Kuriyan et al. revealed the mechanism by which imatinib inhibits the mouse ABL kinase domain[34]. With the extensive use of such targeted drugs, resistance has emerged. The majority of resistant clones are point mutations in the kinase domain of BCR-ABL[18,35]. New generations of drugs, such as dasatinib, bosutinib, and nilotinib, which are more potent than imatinib, were later developed and marketed[36,37]. A specific drug, ponatinib, was developed to target the "gatekeeper" mutation (T315I)[38].

The drug-kinase binding patterns are different and can be divided into seven categories: binding in the ATP-binding pocket in active conformation (type I); binding in the ATP-binding pocket in inactive DFG-in conformation (type I 1/2); binding in the ATP-binding pocket in inactive DFG-out conformation (type II); allosteric binding next to the ATP-site (type III); allosteric binding not next to the ATP-site (type IV); bivalent binding spanning two regions (type V); and covalent binding (type VI)[39]. The inhibitors of ABL mainly fall into two types. Imatinib, nilotinib, bosutinib, and ponatinib are type II inhibitors that binds to the "DFG-out" inactive conformation of ABL kinase. Dasatinib binds to the "DFG-in" active conformation of ABL kinase in the ATP pocket.

Currently, computational methods have been used to understand the mechanisms of clinically identified mutations. However, computational methods that predict drug-resistance mutations in advance are still lacking. In the present study, we report a successful computational and experimental strategy to solving this problem. As BCR-ABL has already accumulated a large number of drug-resistance mutations, and multiple generations of inhibitors are available, this protein offers a good system for testing our computational strategy and experimental validation. The strategy is further applied to EGFR mutations to validate its universality.

## Results

**Computational strategy for de novo prediction of drug-resistance mutations**. Our method employs a GA to simulate the evolution of drug-resistance mutations (EVER). A schematic diagram of the computational strategy is shown in Fig. 1. We first define the residues inside the drug-binding site and translate them back into their respective DNA codons. After random mutation of the DNA sequence, sense mutations are translated into mutations at the amino acid level. Then the structures of the mutated residues are modeled and minimized using a side chain modeling program (the Scap program is used in the current study)[40,41]. The drug molecule is then docked into the mutant structure, and its binding free energy is estimated. The docking program AutoDock Vina is used at this step[42]. We assume that mutations in the binding pocket only affect the direct binding of inhibitors; therefore, no long-range influences are considered. Potential mutants are selected according to the scoring function

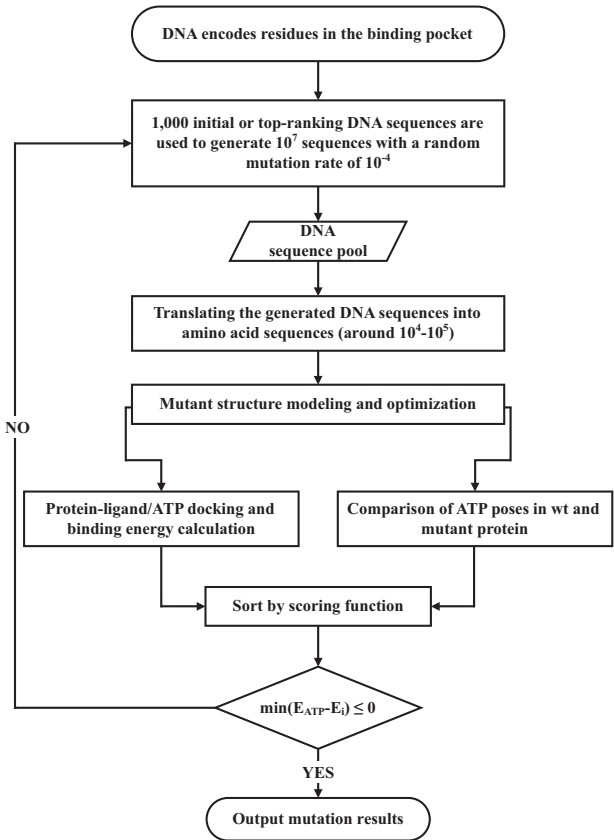

**Fig. 1 Simulation process of EVER based on a mutation-only genetic algorithm to predict drug-resistance mutations.** First, residues inside the drug-binding site are translated back into their respective DNA codons. After random mutation of the DNA sequence, sense mutations are translated into mutations at the amino acid level. Then the structures of the mutated residues are modeled and minimized using a side chain modeling program Scap. The drug molecule is then docked into the mutant structure by AutoDock Vina, and its binding free energy is estimated. Potential mutants are selected according to the scoring function defined in Eq. (1) and subjected to the next round of evolution. The process stops when the binding of the inhibitor to the mutants becomes weaker than its binding to ATP or when the mutations have converged.

(Eq. (1), see below) and subjected to the next round of evolution. The process stops when some of the convergence criteria are reached, including when the binding of the inhibitor to the mutants becomes weaker than its binding to ATP, or when the mutations have converged.

To screen the most probable drug-resistance mutations, we needed to build a suitable scoring function. In general, mutants with weakened drug-binding ability have potential resistance. At the same time, the biological activity of the target protein should be maintained after the mutation. In the ABL system, we docked ATP in the ATP-binding site in the mutant structure and calculated the root mean square deviation (RMSD) of ATP compared to its original position in the wild-type (wt) protein. If the ATP-binding position deviated from the original position with an RMSD greater than 4 Å, the mutation was discarded. The increase in the binding energy of ATP for the target should be within 0.3 kcal/mol. Changes in the binding energy and binding position of ATP were integrated into the scoring function. To prioritize single mutations, the scoring function was normalized to the number of mutations. The scoring function

we used for ABL was defined as

$$\text{Resistance score} = \frac{\Delta E_{\text{mutation}}^{\text{drug}} - \Delta E_{\text{WT}}^{\text{drug}}}{|\Delta E_{\text{mutation}}^{\text{ATP}} - \Delta E_{\text{mutation}}^{\text{drug}}| \cdot \text{RMSD}_{\text{ATP}} \cdot \text{Num}_{\text{mut}}},$$

(1)

where resistance score denotes the drug-resistance potential of the mutation, $\Delta E_{\text{mutation}}^{\text{drug}}$ denotes the binding free energy of the drug for the mutated target, $\Delta E_{\text{WT}}^{\text{drug}}$ denotes the binding free energy of the drug for the wt target, $\Delta E_{\text{mutation}}^{\text{ATP}}$ denotes the binding free energy of ATP for the mutated target, $\text{RMSD}_{\text{ATP}}$ denotes the RMSD of ATP caused by the mutation, and $\text{Num}_{\text{mut}}$ denotes the total number of amino acid mutations.

In most previously reported drug-resistance studies, mutations were directly introduced at the amino acid level to simulate protein mutations. However, this may not reflect actual mutation rates because the codons corresponding to each amino acid have degeneracy. To solve this problem, we performed simulated mutations at the NA level.

In cancer cells, the minimum mutation frequency is estimated to be 0.0042% by sequencing analysis[43]. When cancers enter the middle period, the possibility of drug resistance increases, likely due to the increased frequency of mutations. In the mid-term, the number of cancer cells in the body is estimated to be around $10^{13-14}$, and the number of actively proliferating cells is approximately $10^{8-9}$. The mutation rate of cancer cells entering the middle period is $10^{-5}$ approximately[44,45]. Therefore, in our algorithm, the number of offspring cells containing mutations is expected to be around $10^3$.

As the structural modeling and docking processes are computationally expensive, in our study, the size of the genetic population and the frequency of mutations were reduced to a more computationally manageable level. We first randomly generated $10^3$ gene sequences, with each sequence producing $10^4$ offspring. With a mutation rate of $10^{-4}$, the number of mutations is around $10^3$. For the simulations, we used 50 CPUs (Xeon E5 v2. Core code: Ivy Bridge EP) and each simulation took about 80–90 h.

**EVER reproduces most of the clinically reported BCR-ABL mutations.** We carried out simulations using EVER for the first-generation ABL inhibitor imatinib and the second-generation drugs, nilotinib, and dasatinib. We first checked whether EVER could be used to predict mutations conferring weakened binding strength of the drug to the kinase while preserving the activity of the enzyme by maintaining its ATP-binding energy. The binding energy of ATP for ABL is stable during evolution, as constrained by the scoring function, whereas the binding capacity of the inhibitor for the ABL mutant decays quickly. Taking imatinib as an example, the binding strength of the drug for the target decreases over time (Fig. 2a), whereas the binding energy of ATP for the target remained stable at −7.7 kcal/mol (Fig. 2b).

After the initial test, we then used EVER to predict drug-resistance mutations for imatinib, nilotinib, and dasatinib. A variety of clinical resistance mutations have been discovered after each generation of drugs have been used (Fig. 3 and Supplementary Fig. 1). We compared resistance mutations that are commonly observed in the clinic with those in the top 5% of predicted results. The most commonly observed drug-resistance mutations in the clinic can be found in the predicted results: the distribution of resistance mutations in the clinic is proportional to the predicted results. The most dominant resistance mutation (T315I) accounted for the largest number of predicted results.

In the clinic, common resistance mutations to the first-generation drug imatinib include T315I, Y253H, E255V, G250E,

and Q252H, which account for over 80% of all resistance mutations[46,47]. EVER was used to successfully rediscover these mutations. The three most common resistance mutations, T315I, E255V, and Y253H, represent 57% of the top 5% of mutants, and they also occurred in more than 70% of clinical resistance mutations (Fig. 3a).

The second-generation drugs nilotinib and dasatinib represent a large number of drug-resistance mutations in the clinic, including T315I and Y253H[46,48]. For nilotinib, EVER was used to rediscover the major mutations T315I, Y253H, and E255V, which represent more than 95% of clinical resistance mutations (Fig. 3b). For dasatinib, EVER was used to correctly predict T315I, which occurred in more than 80% of clinically observed drug-resistance mutations. Among the top 5% of predicted mutants, almost all mutations occurred at the T315 site (Fig. 3c).

**Comparison of predicted and clinically observed mutations with predicted and clinically not observed mutations.** We demonstrated that EVER was used to correctly predict most of the clinically observed drug-resistance mutations (p-c mutations). At the same time, EVER was used to identify a number of predicted mutations that have not been clinically observed (p-nc mutations). To develop a reliable method to predict resistance mutations for new generations of drugs, we need to understand why some of the predicted mutations have not been observed clinically. In our simulation, for simplicity, we assumed that the mutations in the drug-binding pocket do not affect protein structure or stability. As some mutants may destabilize the protein, we first checked whether the mutant proteins maintain a thermally stable structure by measuring both denaturation temperatures using nanoDSF[49,50] and secondary structures using circular dichroism[51]. All mutants maintained their original secondary structures with similar denaturation temperatures to the wt enzyme (Supplementary Figs. 2 and 3).

We then measured the binding constants of all the predicted mutants with the corresponding drugs using isothermal titration calorimetry (ITC) or microscale thermophoresis (MST) (Supplementary Table 1 and Supplementary Figs. 4–8). For imatinib, similar to the p-c mutants, all four p-nc mutants (Y253C, V299M, I314V, and F382Y) showed weakened binding strength. For dasatinib, two p-nc mutants (M318I and Y320D) also showed weakened binding. The same phenomenon was observed for the

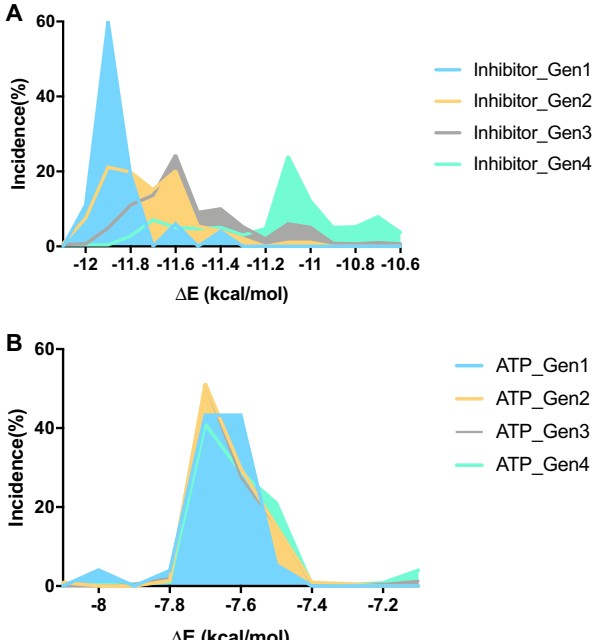

**Fig. 2 Binding energy distribution over times. a** Binding energy distribution of imatinib. **b** Binding energy distribution of ATP. The binding strength of the drug for the target decreases over time (**a**), whereas the binding energy of ATP for the target remained stable at −7.7 kcal/mol (**b**).

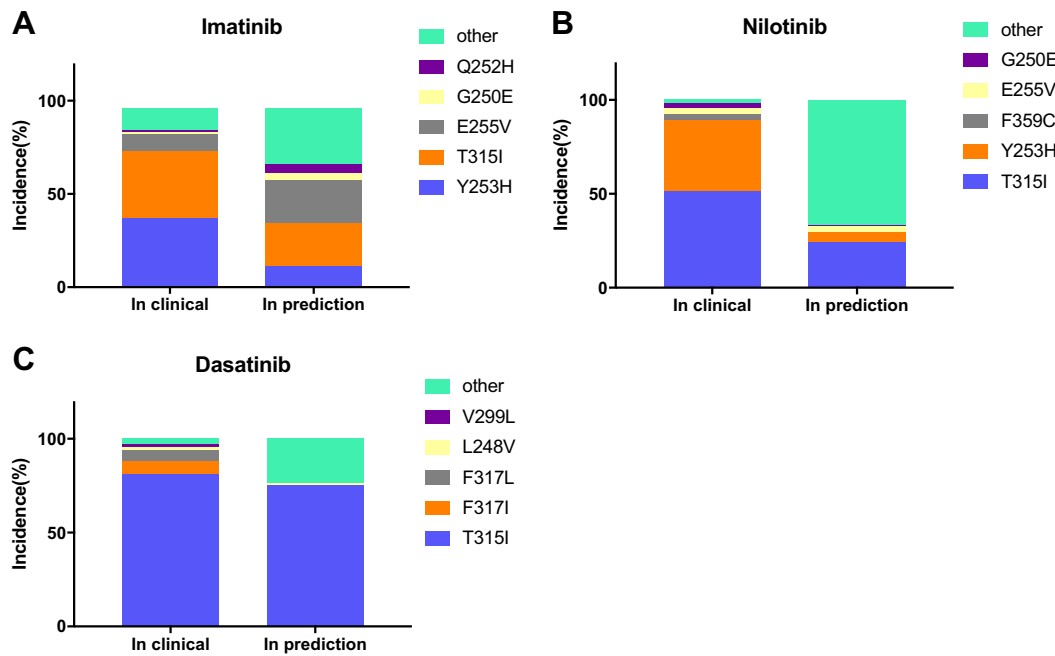

**Fig. 3 Distribution of the most common clinically observed and predicted drug-resistance mutations.** Clinical data are from refs. [25,54,55]. The predicted results only consider the top 5% of drugs developed the last generation. **a** Comparison of the predicted results and commonly observed clinical resistance mutations for imatinib. **b** Comparison of the predicted results and commonly observed clinical resistance mutations for nilotinib. **c** Comparison of the predicted results and commonly observed clinical resistance mutations for dasatinib. $n = 3$ independent samples.

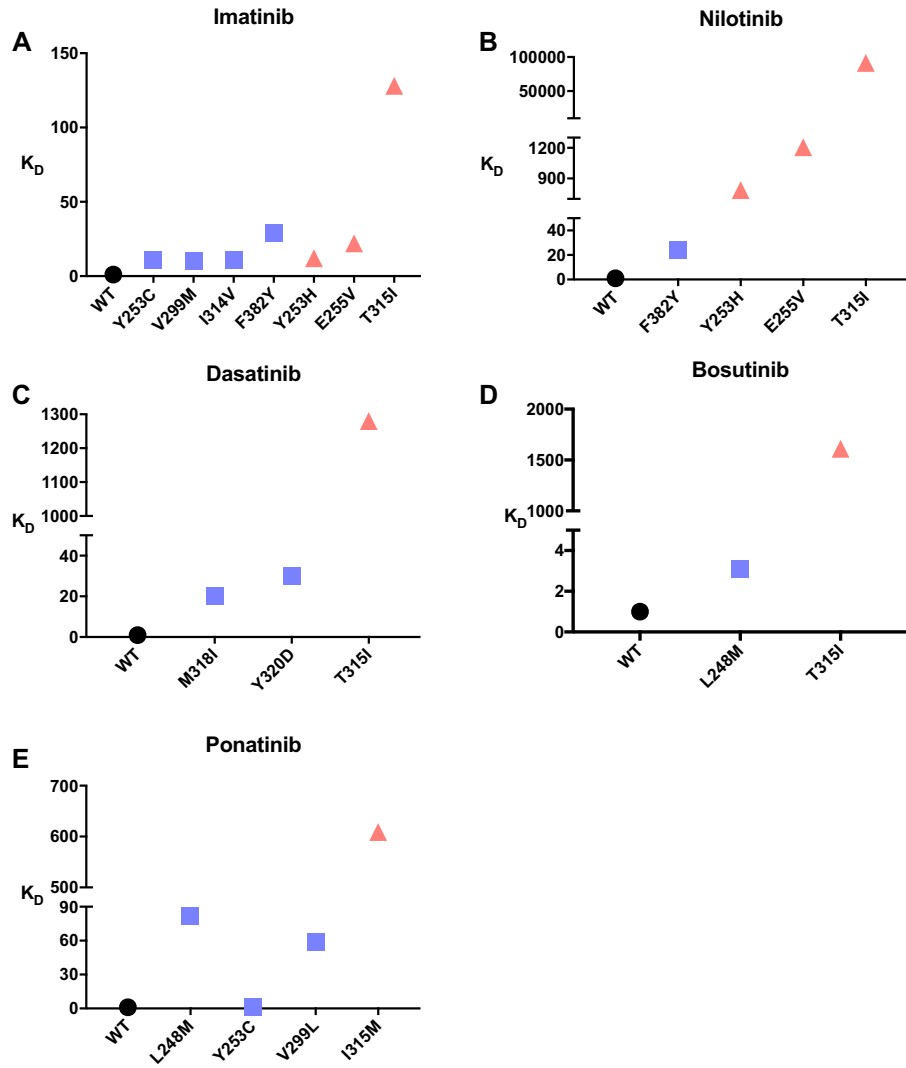

**Fig. 4 $K_D$ measurement of drugs and ABL kinase/mutants.** The $K_D$ of wild-type ABL is normalized. Most clinically resistant mutants (red) have higher $K_D$ values. The non-clinical resistance mutation (blue) $K_D$ is generally between wild-type (black) and clinically resistant mutants, but there are exceptions, such as the F382Y mutation in imatinib's prediction. **a–e** The normalized $K_D$ values of imatinib, nilotinib, dasatinib, bosutinib and ponatinib, respectively. ($K_D$ values for WT are all set to be 1). $n = 3$ biologically independent experiments.

nilotinib p-nc mutant F382Y. These findings confirm that weakened drug binding alone is insufficient to identify clinically observed resistance mutations (Fig. 4).

We further measured the half-maximal inhibitory concentration (IC$_{50}$) values of the three drugs against the wt and mutant ABL proteins. The inhibitory abilities of the drugs against the corresponding p-c mutants were weakened with an increase in the IC$_{50}$ of at least 60-fold. However, the p-nc mutants did not cause significant change in the IC$_{50}$ (<15-fold) (Fig. 5, Supplementary Table 1 and Supplementary Figs. 9–13). Considering that the effective safe plasma concentrations for imatinib, nilotinib, and dasatinib are 1000–3000 ng mL$^{-1}$, <1500 ng mL$^{-1}$, and 2.5–50 ng mL$^{-1}$ [52–54], respectively, the p-nc mutants can all be inhibited by increasing the drug concentration within the safety window.

Inhibitory activity depends not only on the binding of the drug to the target enzyme but also on enzyme catalysis. We experimentally measured enzymatic parameters (Supplementary Table 1). The catalytic efficacy ($k_{cat}/K_{M(ATP)}$) of all clinically observed resistance mutations tested increased less than fourfolds (Y253H, E255V, and T315I). Most of the mutations had a higher

$k_{cat}$ and a slightly altered $K_{M(ATP)}$ compared to wt. In contrast, the p-nc mutations showed broad catalytic efficacy. Among the eight mutants tested, four of them had higher catalytic efficacy by more than sixfold compared to wt (Fig. 6 and Supplementary Figs. 14 and 15), which may not be tolerated by cells.

Based on these computational and experimental results, we propose a stepwise approach for identifying drug-resistance mutations: Step 1: using EVER to identify potential resistance mutations; Step 2: experimentally testing whether the predicted mutant proteins are stable and have catalytic efficacy similar to the wt enzyme; Step 3: measuring the IC$_{50}$ values of the drugs against the mutants from step 2 and compare with their safe dose windows. If the IC$_{50}$ is higher than the highest safe dose, than the mutant is most likely drug-resistant.

**Prediction of bosutinib and ponatinib resistance mutations.** We also predicted resistance mutations for bosutinib and ponatinib (Fig. 7). Bosutinib is a second-generation drug similar to nilotinib and dasatinib that can fight a variety of BCR-ABL resistance mutations in addition to T315I. Ponatinib is the only third-generation BCL-ABL drug currently clinically available.

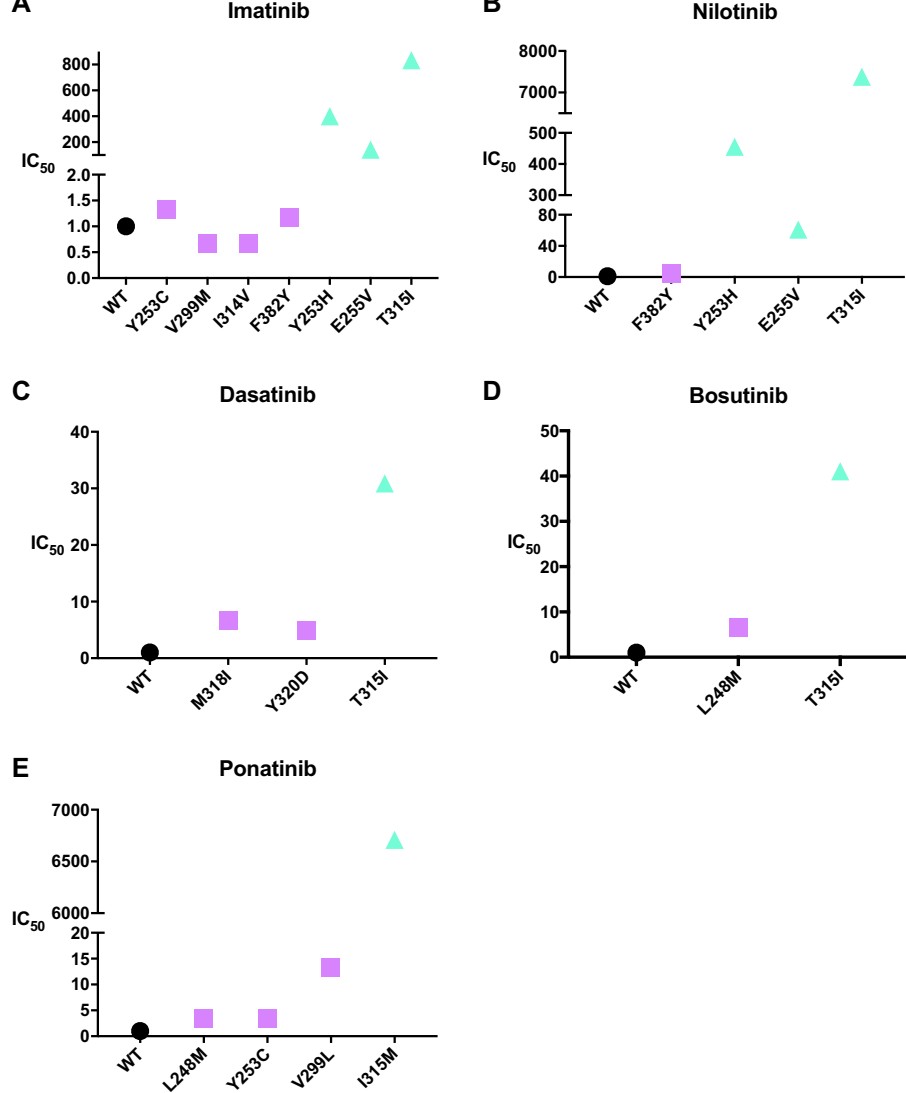

**Fig. 5 IC$_{50}$ measurement of drugs and ABL kinase/mutants.** The IC$_{50}$ of wild-type ABL is normalized. Most clinically resistant mutants (green) have higher IC$_{50}$ values. The non-clinical resistance mutation (purple) $K_D$ is generally between wild-type (black) and clinically resistant mutants. **a–e** The normalized IC$_{50}$ values of imatinib, nilotinib, dasatinib, bosutinib, and ponatinib, respectively. (IC$_{50}$ values for WT are all set to be 1) $n = 3$ biologically independent experiments.

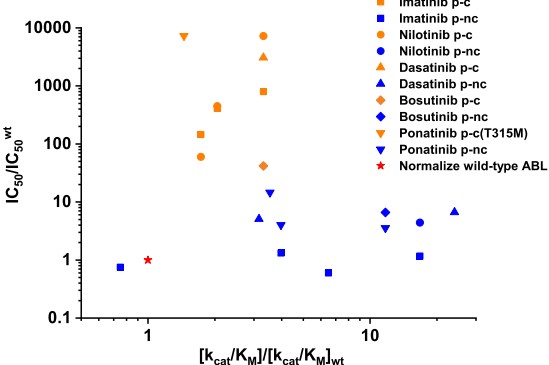

**Fig. 6 Half-maximal inhibitory concentrations and enzyme kinetic constants of ABL mutants and drugs.** All data are normalized to the catalytic efficacy of wt ABL and the corresponding half-maximal inhibitory drug concentrations (red star). The drugs generally have high IC$_{50}$ values against the p-c mutants (orange), while the IC$_{50}$ values against the p-nc mutants (blue) are similar to the wt enzymes.

Although ponatinib is known to be associated with severe adverse vascular events, it is still an approved therapy[55].

Bosutinib works on a large number of single-point resistance mutations induced by the first-generation drugs in the absence of the gatekeeper mutation T315I. The in vitro inhibitory activity of bosutinib is in the sub-nanomolar range[56]. The clinical dosage is between 100 and 500 mg/day, corresponding to a plasma concentration between 31.4 and 150 ng mL$^{-1}$ [57]. We carried out simulation using EVER with the wt kinase for bosutinib. In the prediction results, we found a clinically observed L248V drug-resistance mutation and the T315I gatekeeper mutation (Fig. 7a). Subsequently, we performed $K_D$ and IC$_{50}$ measurements on bosutinib using WT and mutants of L248M (p-nc), T315I (p-c). Both of them follow the rules that we found from imatinib, nilotinib, and dasatinib studies (Supplementary Figs. 8 and 13).

Ponatinib works on all single mutations induced in response to the first- and second-generation drugs, and is recommended for treating patients with the T315I mutation[58]. The in vitro inhibitory activity of ponatinib is in the sub-nanomolar range. The clinical dosage is between 15 and 45 mg/day, corresponding

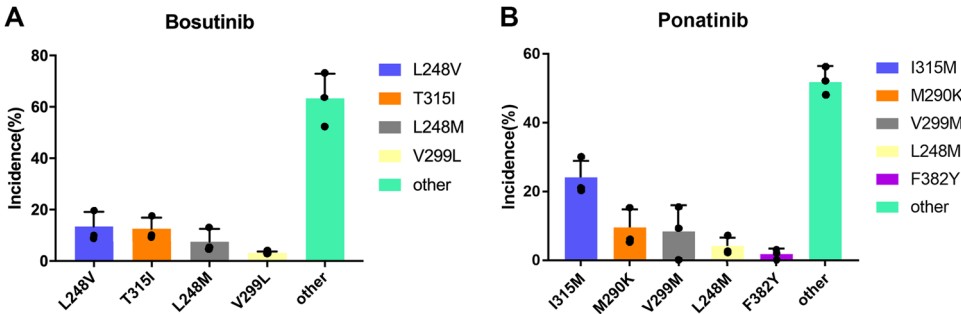

**Fig. 7 Distribution of predicted bosutinib resistance mutations for patients with wt ABL and of ponatinib resistance mutations for patients with the T315I mutation. a** L248V is a relatively unique clinical resistance mutation, and T315I is the gatekeeper mutation. **b** M290K/V299M are usually accompanied by multiple site mutations. Most mutations in the "other" type only occurred once in the three independent runs. $n = 3$ biologically independent experiments.

to a plasma concentration of between 14 and 110 ng mL$^{-1}$ [55]. Unlike the first- and second-generation drugs, no single strong resistance mutations in the kinase domain that directly evolved from the wt enzyme were found after ponatinib treatment. Most of the reported compound mutations include T315I (as one site) or mutations that developed from T315I (e.g., I315M)[59]. Molecular dynamics showed that the binding of ponatinib to ABL causes conformational changes[60]. It was also suggested that the binding of ponatinib to ABL may be driven by entropic changes[61], making ponatinib a difficult case for EVER, as it is unable to handle large conformational changes (Fig. 7b). We carried out two EVER simulations—one for the wt kinase and the other for the T315I mutant—to test whether EVER could be used to yield meaningful results for ponatinib.

The complex structures of ponatinib bound to wt BCL-ABL[59] or the T315I mutant[62] (in which the kinase is in the inactive conformation) were used in the simulations. For the simulation with wt protein, after three rounds of evolution, unlike the simulations for imatinib or dasatinib, no dominant mutations developed (Supplementary Fig. 16). This finding is in accordance with the fact that ponatinib works against almost all single mutations derived directly from wt ABL. In the simulation for the T315I mutant, the evolutionary process converged quickly during the second generation. The single-site drug-resistance mutation, I315M, dominated the results. The IC$_{50}$'s of ponatinib to I315M and wt ABL was 4 μM and 0.44 nM, respectively, that is about 7000-fold weakened inhibition of the drug to I315M compared to wt ABL (See Supplementary Table 1 for detailed data). I315M is the only strong single-site mutation resistant to ponatinib in the clinic to date[38] (Fig. 5e). As the mRNA codon for Thr315 in ABL is ACU, two mutations are needed to produce an AUG (Met), which explains why T315M is not easily observed in simulations with the wt kinase. In contrast, only a single mutation will change an AUU (Ile) into an AUG (Met); thus, simulations starting from T315I can easily produce the I315M mutation. As the current version of EVER was designed to search for strong single-site mutations, compound mutations resistant to ponatinib will be studied in the future.

**EVER application in other systems.** We further tested EVER in another kinase system: the tyrosine kinase domain of epidermal growth factor (EGFR). As our current scoring function was designed for kinase inhibitors, for systems other than kinase, the scoring function will need to be tuned. EGFR mutations are the most common drivers of non-small cell lung carcinoma[63]. The main mutations are codon deletions at positions 746–750, changes in the ATP-binding angle, and the L858R point mutation, which enhances kinase activation. EGFR inhibitors were

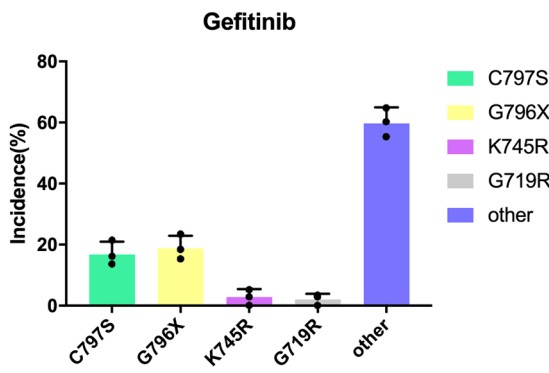

**Fig. 8 Distribution of predicted gefitinib resistance mutations for patients with wt ABL.** C797S, G796C/D/R/S, and G719R are the major mutation types clinically observed. $n = 3$ independent simulations.

developed for the above EGFR mutations, such as gefitinib, afatinib, dacomitinib, and osimertinib. As the last three inhibitors are covalent inhibitors, we chose gefitinib, a reversible competitive inhibitor to test EVER. We used EVER to predict resistance mutations for gefitinib using the same computational settings and parameters as we did for BCR-ABL inhibitors. EVER successfully predicted the common clinically observed resistance mutations of G719R, G796C/D/R/S, and C797S (Fig. 8). This shows that our scoring function is reasonable, and the algorithm has certain universality.

However, the gatekeeper mutation, T790M, in the EGFR system, did not dominate in the results. It was found that the resistance mechanisms triggered by T315I in ABL and T790M in EGFR are different. The T315I mutation blocks the binding of the drug to the target protein by the longer side chain of Ile, while T790M achieves drug resistance by altering the ATP binding affinity and catalytic efficiency of EGFR[64]. As EVER currently cannot handle changes in catalysis, it is understandable that T790M did not show up.

## Discussion

In this study, we have developed a computational algorithm, named EVER, to predict drug-resistance mutations. EVER consists of three main steps—defining the binding-site sequence, modeling, and docking of the protein-inhibitor, and scoring—to simulate the emergence of drug-resistance mutations. For protein targets with the structure of the protein–drug complex available, EVER can be applied to predict potential mutations induced by the drugs. There were previous studies on BCR-ABL drug resistance mutations by experimental screening or by computational

retrospective analysis (summarized in Supplementary Table 2). However, computational methods to predict BCR-ABL drug-resistance mutations in advance were not reported. EVER can de novo predict drug-resistance mutations without any training process, and can be used to predict drug-resistance mutations for newly marketed drugs.

The scoring function can be tailored for different target systems. For protein kinase ATP competitive inhibitors, we designed the scoring function to consider the binding energies of both the drug and ATP to the target, as well as the binding pose of ATP. The rationale of constructing this scoring function is the following: (1) Drug resistance mutations should weaken drug binding. However, only the differnce in drug binding to mutant and wt kinase (the nominator term in the scoring function) cannot guarantee that the mutant kinase remains active (thus be clinically observable). (2) In order to make sure that the kinase remains active after mutation, we introduced the denominator term by considering the difference between ATP and the drug binding, as well as the binding pose change of ATP. We have tried to only use the drug binding energy difference term as the scoring function and many of the top-ranking mutants cannot bind ATP correctly (data not shown).

The goal of evolution is to weaken drug binding while maintaining constant ATP binding. This kind of scoring function design can be generally applied to inhibitors that compete with cofactors for binding. Our results indicate that the scoring function designed works effectively. The computational evolutionary process reached its termination by three of four rounds of mutant generation.

In the ABL system that we tested, EVER was able to reproduce most of the strong single-site mutations for all three generations of drugs that have been used in the clinic. We further carried out experimental studies to understand why some of the predicted mutants were not observed in the clinic and to define the characteristics of the predicted and clinically observed mutations. We found that clinical mutations do not obviously change the activities of the enzymes (within threefold). Predicted mutations that significantly increase the activities of enzymes have not been observed in the clinic, which is consistent with reports that the activities of enzymes in living organisms are strictly regulated[65,66]. We have also found that simply decreasing drug binding strength is not enough to produce drug resistance, as the enzyme activity change also needs to be considered. To ensure that the enzyme is active, the ATP-binding strength must be calculated at the same time. Currently, it is not practical to include computations on enzyme catalysis, as these calculations are too computationally expensive. Our suggestion is to make predictions without considering enzyme catalysis first and then test the predictions experimentally in vitro. In the ABL system, we found that the $IC_{50}$ value is a good predictor of clinical mutants.

The current version of EVER can be used to simulate single-point mutations. For multi-point mutations, scoring function need to be tuned and conformation changes should also be considered. EVER is not suitable for non-competitive inhibitors or allosteric inhibitors[67], or inhibitors targeting downstream signaling nodes[68]. As EVER is based on molecular docking and restricted by the computing power, only the amino acid residues that directly interact with drugs can be predicted and analyzed. Amino acid mutations that are far away from the interaction region cannot be treated. The enzyme catalytic process is also not considered in the current version of EVER. However, with rapid increase of computing power and the development of more rapid and accurate simulation methods for conformation sampling, binding free energy calculations, and enzyme catalysis, we expect more accurate predictions of drug-resistance mutations in the future. Target selectivity of drugs is a considerable issue as many kinase inhibitors are promiscuous. Understanding the regulatory mechanisms of disease-related molecular networks to which the target belongs also provides key information for more accurate predictions of drug-resistance mutations.

## Methods

**Generation of mutants according to a specific mutation ratio.** We used a genetic algorithm (GA) algorithm as the mutation strategy for offspring generation. In the first generation, $10^3$ gene sequences were randomly generated; each sequence produced $10^4$ offspring with a mutation rate of $10^{-4}$. As the structural modeling and docking processes are computationally very expensive, the size of the population and the frequency of mutations were reduced to a computationally manageable level. In the GA algorithm, population size was controlled by deleting the lowest-ranking individuals, allowing only the top 5% of individuals containing mutations. If the eligible sequence number exceeded 1000, then the top 1000 mutant sequences were selected. The original sequences with the highest scores were used to fill the population if the eligible sequence number was less than 1000. The DNA sequence of each individual was translated into a protein sequence, which was then subjected to protein structure modeling and molecular docking. The final evaluation score of a mutant was calculated, according to Eq. (1). The genetic evolution was considered complete when the top-ranked mutant yielded a lower binding affinity to the drug than to ATP.

**Side chain packing calculations.** We used the program Scap (http://honig.c2b2.columbia.edu/scap/) to model the structures of the protein mutants[40,41]. A Perl script was used to call the Scap module, with the aim to convert the mutated residues into the corresponding variation in protein structure. Scap is used to build side chain conformations using its coordinate rotamer libraries. As we do not want the mutation to change the protein structure significantly, we chose an AMBER force field with a heavy atom model and a mixed side chain rotamer library. The other parameters were set to the default values, which allowed a relatively stable mutant protein conformation. We used the Scap program to generate structures of thousands of residue mutations.

**Protein-drug docking.** We used the AutoDock Vina program (http://vina.scripps.edu/index.html, version 1.1.2) to build the drug-mutant and ATP-mutant structures and calculate the binding scores[69]. Default AutoDock Vina parameters were used. The superposition module in Schrödinger[70] was used to calculate the RMSD of ATP in the crystal structure and in the mutants.

**ABL structures.** During the simulation, we used different ABL crystal structures for different compounds. The structure of ABL complexed with ATP was derived from the ATP-peptide conjugate complex (PDB code: 2G1T) with the protein in an inactive conformation[71]. The structure used for imatinib was 2HYY with the complexed protein in an inactive conformation[72]. The structure used for nilotinib was 3CS9 with the complexed protein in an inactive conformation[73]. For dasatinib, we used 2GQG with the complexed protein in an active conformation[48]. Ponatinib was designed for the T315I mutant. It avoids steric hindrance with the side chain of isoleucine. 3IK3 (with a T315 mutation) complexed with ponatinib was also used. The mutant protein 3IK3 adopts an inactive conformation[62]. The wide-type protein structure in this case was obtained by mutating I315 back to T315.

**EGFR structures.** The structure of EGFR tyrosine kinase domain in complex with ATP was derived from the thiophosphoric acid O-[(adenosyl-phospho)phosphor]-S-acetamidyl-diester complex (PDB code: 2GS6) with the protein in an active conformation[74]. The structure used for gefitinib was 4I22 with the complexed protein in an active conformation[75].

**Compounds and substrates.** Imatinib (MW493.6), nilotinib (MW529.52), dasatinib (MW488.01), and ponatinib (MW532.56) were purchased from Selleck (https://www.selleck.cn/). The substrate peptide was obtained from GL Biochem (Shanghai) Ltd. and had a sequence of Lys-Lys-Gly-Glu-Ala-Ile-Tyr-Ala-Ala-Pro-Phe-Ala-NH2 (Directory peptide Cat # 86721).

**Expression and purification of ABL and mutants.** The kinase domains of human c-ABL (residues 222–500, NM_005157.5) were subcloned into the *Nde*I and *Xho*I restriction sites of the pET-28a vector[76]. The plasmids were transformed into *Escherichia coli* BL21 (DE3) cells, plated on LB agar containing kanamycin (50 μg mL$^{-1}$), and grown overnight at 37 °C. The next day, the colonies from the plates were resuspended in expression media (LB agar containing kanamycin, 50 μg mL$^{-1}$). Cultures were grown to an $OD_{600}$ of 1.2 at 37 °C and cooled for 1 h with shaking at 16 °C prior to induction for 22 h at 16 °C with 0.1 mM IPTG. Cells were harvested by centrifugation at $7000 \times g$ at 4 °C for 15 min and stored at −80 °C. The bacterial pellet was resuspended in Buffer A [50 mM Tris (pH 8.0), 500 mM NaCl, 20 mM imidazole] for immediate purification using nickel ion affinity chromatography. Elution of the protein from the column was achieved using a Buffer B gradient [50 mM Tris, 500 mM NaCl, 500 mM imidazole

(pH 8.0)], which increased from 0% to 100% over 15 min. The eluted protein was concentrated to 2 mL and subjected to gel filtration using an S200 column[77,78]. The buffer used for the gel filtration was Buffer C [50 mM Tris, 500 mM NaCl (pH 8.0)]. The protein was eluted at 55 min. The purity of the protein was verified by SDS-PAGE.

For the mutants, primers were designed based on the predicted mutations and synthesized by GENEWIZ. Mutations were made using the Fast Site-Directed Mutagenesis kit from TIANGEN, according to the manufacturer's instructions, and verified by DNA sequencing. The mutant proteins were expressed and purified following the expression and purification of the wt protein.

**In vitro kinase inhibition assay**. To assess the ability of the drug to inhibit the wt and mutant kinases, we used the ADP-Glo Kinase Assay kit from Promega[79], which measures the amount of ADP produced in the reaction. The inhibition rates of the drugs at different concentrations were calculated by comparing the amount of ADP produced with and without the drug. Data were analyzed using the Hill1 model in the OriginLab2018 software package.

**In vitro enzyme activity assay**. ITC experiments were carried out using an ITC200 instrument (Microcal Inc.). ITC has been demonstrated to directly measure the kinetics and thermodynamic parameters ($k_{cat}$, $K_M$, $\Delta H$) of enzymatic reactions[80]. A one-step method was used to measure the enzymatic parameters. The BCR-ABL concentration was in the nanomolar range, with the ATP/substrate concentration at least three orders of magnitude higher than the enzyme concentration and above the $K_M$ [BCR-ABL (10 nM) and substrate (1 mM) in the cell, and ATP (1 mM) in the syringe].

The thermal change ($Q$) is proportional to the reaction enthalpy ($\Delta H$) and the number of moles of product ($n$), whereas the moles of product equal the total volume ($V$) multiplied by the concentration [$P$]:

$$Q = n \cdot \Delta H = V \cdot [P] \cdot \Delta H. \quad (2)$$

According to Eq. (3), the $\Delta H$ of the reaction can be obtained by integrating the curve of the Method 1 experiment

$$\Delta H = \frac{\int_{t=0}^{\infty} \frac{dQ}{dt} dt}{V[S]_{t=0}}. \quad (3)$$

The rate of product formation ($dP/dt$) is related to the heat ($dQ/dt$) generated at the same time by the following relationship

$$v = \frac{dP}{dt} = \frac{1}{V \cdot \Delta H} \cdot \frac{dQ}{dt}. \quad (4)$$

Then the data were fitted to the Michaelis–Menten equation. Data were analyzed using the enzyme activity model in the Origin software package provided with the instrument.

**Quantitative measurement of binding constants for kinase mutants and inhibitors**. ITC and MST were used to measure the binding constants of the drugs to the protein targets[81,82]. The binding constants of the drugs to ABL and its mutants were measured using ITC whenever possible. ITC experiments were carried out using an ITC200 instrument (Microcal Inc.). Both the final drug solution and protein solution contained 2% DMSO. ITC measurements were performed at 30 °C in 20 mM MES (pH 6.4), 50 mM NaCl, and 2% DMSO. We diluted the drug solution (400–500 μM) into the protein solution 10-fold. Data were analyzed using the single binding-site model in the Origin software package provided with the instrument.

As the binding of nilotinib and ponatinib to ABL produced nondetectable levels of heat, their binding was measured using MST[83,84]. MST experiments were carried out using a Monolith NT.115 system (nanoTEMPER) in phosphate-buffered saline buffer. The His-tagged dye was incubated with the protein for 30 min, and the mixture was then centrifuged at $15,000 \times g$ for 10 min at 4 °C. Different drug concentrations were mixed with the proteins at a 1:1 (v/v) ratio and transferred to Monolith NT.115 capillaries. Data were analyzed using MO.Affinity analysis software.

**Statistics and reproducibility**. The computational simulation for each drug was repeated independently for three times. The top 5% ranked mutants were selected for statistic analysis. The protein denaturation temperature, secondary structure, binding constant ($K_D$), and half-inhibition concentration ($IC_{50}$) data were all measured independently for three times.

**Reporting summary**. Further information on research design is available in the Nature Research Reporting Summary linked to this article.

## Data availability

All data are available in the main and supplementary files. Source data are available in Supplementary Data 1. Data for all figures, Supplementary figures, and computational output, can also be downloaded at https://github.com/PKUMDL-AI/EVER/tree/master/data/ .

## Code availability

The Scap program is available from:http://honig.c2b2.columbia.edu/scap/. The AutoDock Vina program is available from http://vina.scripps.edu/index.html, version 1.1.2 The code for EVER algorithm is available from https://github.com/PKUMDL-AI/EVER/tree/master/code/.

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

## Acknowledgements

This work was supported in part by the Ministry of Science and Technology of China (2016YFA0502300 and 2015CB910300) and the National Natural Science Foundation of China (21633001 and 21673010). The computational work was performed using Peking University's "Polaris" high-performance computing platform. We thank Dr. Youjun Xu, Dr. Weilin Zhang, and Bo Deng for their help in computer programming and data processing.

## Author contributions

J.P. and L.L. conceived the project. J.L., J.P. and L.L. designed the experiments. J.L. performed the experiments. J.L., J.P. and L.L. analyzed the results and wrote the paper.

## Competing interests

The authors declare no competing interests.
