## [Peer Review File · Communications Biology]

Reviewers' comments:

Reviewer #1 (Remarks to the Author):

A computational pipeline for prediction of mutations conferring resistance to small-molecules is developed. This pipeline incorporates analysis of drug binding site DNA sequence evolution, modeling of impact on protein structure and drug binding, and a scoring and selection process. This pipeline is applied to the ABL kinase domain to predict mutations conferring resistance to ABL kinase inhibitors in CML. It appears to predict the most common resistance mutations to imatinib, nilotinib, dasatinib, and ponatinib with reasonable fidelity to the frequency with which these mutations are observed in the clinic.

1. Since there are some additional ABL inhibitors now in clinical use (e.g. bosutinib), it would be useful to expand to these other drugs.
2. The comparison of predicted ponatinib mutational frequency with that actually observed in the clinic was not found by this reviewer.
3. For all 4 of the drugs studied, extensive laboratory work has been conducted to predict resistance mutations. It would be useful to include a summary of these results in the figures for comparison to see how well each strategy is able to predict clinically impactful mutations.
4. While it is useful to see this for ABL, this is also an area that has been already highly studied. It would be important to expand this to other genes, perhaps even some non-kinase targets (e.g. BCL2).
5. The text alludes to an experimental study in which ABL I315M is 7,000 fold less sensitive to ponatinib, but this reviewer could not find the results of that experiment shown in the paper.

Reviewer #2 (Remarks to the Author):

The paper revolves around predicting and identifying which mutations in a protein can make it resistant to an ATP-competitive inhibitor. To this end, the authors combine a genetic algorithm with side-chain conformational prediction and protein-ligand docking simulations. For several drugs targeting Bcr-Abl, they find a good overlap of predicted mutations and those that are observed. They subsequently follow-up experimentally on the mutations were not predicted correctly.

Overall, this is an interesting approach, and the methodology and results are clearly explained. The follow-up experiments including target inhibition and binding parameters provide useful insights in where the benefits and limitations of the simulations lie. I do however have two major concerns which I think should be addressed before the manuscript can be published.

1. It is unclear how this approach relates to existing publications, in particular [1] and [2]. The present manuscript does mention relevant literature in the introduction and also references [1], but they essentially sum up the existing approaches with "However, the prediction performance was closely related to the training sets." It is not clear to me what is meant with this sentence, and how the present approach differs. Further, if previous methods were limited in that they could not predict outside of a training set, then I would expect that the present paper does predict outside of the training set (i.e. make predictions for another protein than Bcr/Abl). However, the present paper also only makes predictions for one target. Paper [2] is not referenced but does seem quite relevant at first glance.
2. As with any algorithm & simulation, there are various settings and parameters. It is unclear to what extent these settings and parameters were varied during the development of the methodology. For

example, was the presented scoring function the only one that was tried, or were more scoring functions attempted? For the parameters, especially the parameter controlling which percentage of mutations to keep in the final prediction (5%) could have a major influence on how well the predictions match the clinical observations. Was this threshold varied?

In other words, were all settings (the scoring function, thresholds, etc) chosen at the outset and then kept fixed before looking at the overlap with clinical observations? Or were the settings modified as the project progressed? In the former case, then this should be indicated in the methods. In the latter case, there is the potential for overtraining, and the accuracy of a new prediction (a new drug target) would be needed to be able to justify whether the approach can truly predict.

Finally, the text contains quite a few grammatical errors. The paper is still clearly comprehensible, and I realize the authors are non-native speakers. Nevertheless it makes the paper uncomfortable to read and gives an impression of sloppiness (which I don't think is justified). Perhaps the authors can find a native speaker who would be willing to provide some editing work?

Some examples are listed below, but I stopped annotating after the first pages:

p3 line 50 affect->affecting

p3 line 53. Do you mean "blockbuster"?

p4 line 66 "part of the effort has been to"

p4 line 80 "protease that interacted"

Minor points:

- For the overlap with clinical mutations, the authors should provide the following information: exactly for which part of the protein were predictions made? And were all clinical mutations in the entire protein included, or only those mutations which occur inside the region for which predictions were made?

- Figure 4; the axis labels are incorrect. The graph does not depict "IC50", but the ratio between the IC50 and that of the corresponding wild-type ABL; please make sure this is indicated on the axis label

- I think it would be good to include a summary of supplementary figures 8-11 as a main figure, it is an important result - perhaps as a bar graph of IC50s?

- Supplementary figures 6 and 7 compare measurements for different mutations; but the scale of the axes is (slightly) different between the graphs. This makes it difficult to directly compare the data.

Can the authors make all axes exactly the same scale?

- The same goes for supplementary figures 8-11.

- The computation time is mentioned as a limiting factor. Can the authors make a brief mention somewhere of the typical computation time involved (e.g. number of cores used and hours)?

- At the end of page 15, the authors propose a strategy for finding new drug resistant mutations.

Based on Figure 4; it seems to me that step #2 is not necessary, as only the IC50 can already distinguish p-c from p-nc mutations. Is that indeed correct, or do I miss something? It is of course always good to check the k_{cat}/K_M , and fig 4 provides useful information for the manuscript, but the k_{cat}/K_M does not seem to add information for classifying observed vs non observed mutations.

- Supplementary Figure 1 mentions that the results were of statistical convergence. How was this tested?

References:

[1] Hou T, Zhang W, Wang J, Wang W. Predicting drug resistance of the HIV-1 protease using molecular interaction energy components. *Proteins*, 2009

[2] Kevin Hauser, Christopher Negron, Steven K. Albanese, Soumya Ray, Thomas Steinbrecher, Robert Abel, John D. Chodera & Lingle Wang. Predicting resistance of clinical Abl mutations to targeted kinase inhibitors using alchemical free-energy calculations. *Communications Biology*, 2018.

Reviewer #3 (Remarks to the Author):

The authors demonstrate an algorithm to identify single point mutations of the ABL kinase domain that can predict drug resistance. Such an algorithm could contribute to drug design and is therefore worthy of consideration by researchers involved in identifying drugs to combat resistance to BCR-ABL inhibitors and potentially to drugs designed to circumvent resistance to other protein kinases. Specific points (not in any particular order):

- 1) Abstract. The claim that EVER might be of value to predict efficacy outside of a drugs safety window cannot be supported, since such a prediction would need prior information about target selectivity, preclinical animal testing and phase 1 clinical data.
- 2) The Introduction should be redrafted to be more concise and focus more on BCR-ABL inhibitors to treat CML (e.g. HIV is not relevant to the current study). The need for target selectivity of drugs should be mentioned.
- 3) The landmark study of imatinib resistance by Daley should be cited and discussed (Cell 2003;112:831).
- 4) The structural biology of ABL (active & inactive conformations) should be discussed, together with their relevance and extrapolation to BCR-ABL.
- 5) The strategy to combat mutations by non ATP-competitive approaches (e.g. J Med Chem 2018;61:8120) or by targeting down stream signalling nodes should be mentioned.
- 6) The Results section includes much discussion (e.g. lines 193-213) which should be moved to the Discussion section. The assumption on line 127 is a major limitation, since mutations or detected outside of the ATP-binding site, as the situation with ponatinib (line 292), and these warrant discussion.
- 7) The comment on line 234 that mutations destabilise the protein is incorrect, since a destabilised protein will not drive disease. Mutants must be able to bind ATP and be functional in phosphorylating their substrates: the mutations destabilise particular protein conformation and this need to be discussed in detail (See Mini Rev Med Chem 2004;4:285).
- 8) Decreasing drug binding/potency can promote resistance, as shown nicely by Duyster et al (e.g. Fig 1 of Blood 2006;108:1328); the sentence (line 339) needs to be redrafted.
- 9) On line 337 the authors write that dasatinib can bind DFG-in & DFG-out ABL, but this is based on speculation and has been shown not to be the case experimentally (J Biol Chem 2008;283:18292) and this needs to have more discussion.
- 10) In the case of ponatinib drug-resistant compound mutations might be mentioned (see Blood 2016;127:703), where the authors could discuss the potential of EVER to predict these.

Some specific recommendations to text:

Line 22: ...clinically resistant...

Line 23: ...develop drugs to combat resistance....

Line 33: ...although...

Line 47: ...the 2050s. Developing drugs to combat resistance in cancer, such...

Line 51: In CML point mutations are the most common mechanism of resistance

Line 53: ...including important drugs such as...The authors should consistently use International Nonproprietary Names throughout, and these should not be capitalised.

Line 57: ...a solution...

Line 58: Mutations resistant to imatinib were predicted in advance by Corbin et al. Blood 2000;96:470a (Abstract#2025) & J Biol Chem 2002;277:32214.

Line 61: ...hindrance...

Line 64: Classify as intrinsic and extrinsic mechanisms.

Line 68: PCR does not generate mutations!

Line 102: ..protein that is not autoregulated and is always....

Line 105: ..inhibits the kinase activity of ABL.

Line 106: The authors should note that this was using mouse ABL. Do the authors use mouse or human ABL for their study

Line 107: ..clones harbour point...

Line 255: The steady-state plasma level of 400 mg BID nilotinib is not as written and this should be changed (see Eur J Clin Pharmacol 2012;68:723.

Line 281: ..for treating refractory CML...

Point-to-point response to the referees' comments:

Reviewers' comments:

Reviewer #1 (Remarks to the Author):

A computational pipeline for prediction of mutations conferring resistance to small-molecules is developed. This pipeline incorporates analysis of drug binding site DNA sequence evolution, modeling of impact on protein structure and drug binding, and a scoring and selection process. This pipeline is applied to the ABL kinase domain to predict mutations conferring resistance to ABL kinase inhibitors in CML. It appears to predict the most common resistance mutations to imatinib, nilotinib, dasatinib, and ponatinib with reasonable fidelity to the frequency with which these mutations are observed in the clinic.

1. Since there are some additional ABL inhibitors now in clinical use (e.g. bosutinib), it would be useful to expand to these other drugs.

Response: Following this suggestion, we further tested our EVER algorithm on Bosutinib and carried out wet experiments to validate the EVER computation results (Now all the 5 clinically used inhibitors mainly targeting ABL were tested). Computational results on bosutinib also correlated well to the experimental results. Results and Discussion on these were added in the revised manuscript. (Figure 4, 5, 6 and 7 were modified accordingly and Figure S8 and S13 were added)

2. The comparison of predicted ponatinib mutational frequency with that actually observed in the clinic was not found by this reviewer.

Response: This is because resistant mutations to Ponatinib are usually multi-point mutations and only I315M is the prominent single-point resistant mutation. It is not possible to generate the probability single point drug-resistant mutations in the clinic. We only did the comparison for the first 3 drugs (imatinib, Nicotinib and Dasatinib), for which single-point resistant mutation are predominant in the clinic.

3. For all 4 of the drugs studied, extensive laboratory work has been conducted to predict resistance mutations. It would be useful to include a summary of these results in the figures for comparison to see how well each strategy is able to predict clinically impactful mutations.

Response: There were previous studies on BCR-ABL drug resistant mutations by experimental mutagenesis screen or by computational retrospective analysis. We have added a summary table for these studies in Supplementary Table 2. However, previous computational methods have only been used to analyze why the clinic mutations are drug resistant, but not to *ab initio* predict BCR-ABL drug-resistance mutations. Our algorithm can computationally *de novo* predict drug-resistant mutations with only the kinase structure and the drug molecule information without any training process. We tested the validity of the method by recapture the clinically observed drug-resistant mutations and discussed its potential application in predicting possible mutations for new generation of drugs. We have made this clear in the revised manuscript. (line 398-406).

4. While it is useful to see this for ABL, this is also an area that has been already highly studied. It would be important to expand this to other genes, perhaps even some non-kinase targets (e.g. BCL2).

Response: We thank the reviewer for the suggestive comment. As currently our scoring function is specifically customized to kinases (Our scoring function contains items related to ATP), the current version of EVER can be only used for kinase inhibitors. (Of course it can be used for other targets with scoring function tuning in the future) . As a response to this comment, we have applied the algorithm to another kinase target (EGFR), without any setting and parameter change, and also got good predictive results. Please see details of the expansion work in the revised manuscript (A new Figure 8 was added in the revised manuscript).

5. The text alludes to an experimental study in which ABL I315M is 7,000 fold less sensitive to ponatinib, but this reviewer could not find the results of that experiment shown in the paper.

Response: In our experiments, ponatinib inhibited wild-type and I315M ABL with IC_{50} of 0.44 nM, 4 μ M, respectively, indicating that ABL I315M is about 7,000 fold less sensitive to pontatinib. Please refer to these data in Supplementary Table 1. We have made this clear in the revised manuscript (line 344-347).

Reviewer #2 (Remarks to the Author):

The paper revolves around predicting and identifying which mutations in a protein can make it resistant to an ATP-competitive inhibitor. To this end, the authors combine a genetic algorithm with side-chain conformational prediction and protein-ligand docking simulations. For several drugs targeting BCR-ABL, they find a good overlap of predicted mutations and those that are observed. They subsequently follow-up experimentally on the mutations were not predicted correctly.

Overall, this is an interesting approach, and the methodology and results are clearly explained. The follow-up experiments including target inhibition and binding parameters provide useful insights in where the benefits and limitations of the simulations lie. I do however have two major concerns which I think should be addressed before the manuscript can be published.

1. It is unclear how this approach relates to existing publications, in particular [1] and [2]. The present manuscript does mention relevant literature in the introduction and also references [1], but they essentially sum up the existing approaches with "However, the prediction performance was closely related to the training sets." It is not clear to me what is meant with this sentence, and how the present approach differs. Further, if previous methods were limited in that they could not predict outside of a training set, then I would expect that the present paper does predict outside of the training set (i.e. make predictions for another protein than BCR/ABL). However, the present paper also only makes predictions for one target. Paper [2] is not referenced but does seem quite relevant at first glance.

Response: In the introduction section, we reviewed several kind of methods for drug-resistance mutation computaion. We have added reference [1] (reference 28 in the revised manuscript) and reference [2] (reference 25 in the revised manuscript) now and added more introduction to these methods. We are sorry for the mis-leading statement about "However, the prediction performance was closely related to the training sets". This sentence only refers to "Methods using sequence information and machine learning" and have no relationship to other reference like [1] and [2]. Yes it is ture that paper [2] is quite relevent. It is quite new and we only found it after we had already submitted our manuscript. In the work of Paper [2], through FEP, the authors are possible to accurately calculate the binding energy change after mutations and then correctly classify mutations as resistant or susceptible. Our method is a quite different one that it can *de novo* predict drug-resistant mutations before they occur or being clinically observed. We have applied the algorithm to another target (EGFR) and, once again, got good predictive results (see in section "EVER application in other systems" in the revised manuscript).

2. As with any algorithm & simulation, there are various settings and parameters. It is unclear to what extent these settings and parameters were varied during the development of the methodology. For example, was the presented scoring function the only one that was tried, or were more scoring functions attempted? For the parameters, especially the parameter controlling which percentage of mutations to keep in the final prediction (5%) could have a major influence on how well the predictions match the clinical observations. Was this threshold varied?

In other words, were all settings (the scoring function, thresholds, etc) chosen at the outset and then kept fixed before looking at the overlap with clinical observations? Or were the settings modified as the project progressed? In the former case, then this should be indicated in the methods. In the latter case, there is the potential for overtraining, and the accuracy of a new prediction (a new drug target) would be needed to be able to justify whether the approach can truly predict.

Response: Thanks to the comment. As can be seen from the formula of our scoring function (eq. 1 below), no tunable parameters were used. Our rationale of formulating this scoring function is: (1) drug resistance mutations should weaken drug binding, which is reflected in the numerator term. The difference in binding energy has been used in previous studies. However, only using this term cannot guarantee that the mutant kinase remains active (thus be clinically observable). (2) in order to make sure that the kinase remains active after mutation, we introduced the denominator term by considering the difference between ATP and the drug binding energy and the binding pose change of ATP. We did try first by only using the difference of drug binding energy between wild-type and mutant enzyme, and many of the resulting mutants have significantly reduced ATP binding and/or wrong ATP orientation. That is why we used the current form of scoring function. We have made this clear in the revised manuscript both in the Results and Discussion sections.(line 144-160; line 410-418)

For the GA algorithm, the results were usually not sensitive to the settings and parameters. We choose the best settings and parameters (including the percentage of mutations to keep (5%) in the final prediction) of GA from the pre-test of the ABL-imatinib simulation. In the later simulations for nilotinib, dasatinib, bosutinib and ponatinib, our parameters are no longer changed. Even for the application to another target EGFR we did not change the settings and parameters. Satisfactory prediction results were obtained by the same scoring function, settings and parameters.

$$resistance\ score = \frac{\Delta E_{mutation}^{drug} - \Delta E_{WT}^{drug}}{|\Delta E_{mutation}^{ATP} - \Delta E_{mutation}^{drug}| \cdot RMSD_{ATP} \cdot Num_{mut}} \quad (eq. 1)$$

Finally, the text contains quite a few grammatical errors. The paper is still clearly comprehensible, and I realize the authors are non-native speakers. Nevertheless it makes the paper uncomfortable to read and gives an impression of sloppiness (which I don't think is justified). Perhaps the authors can find a native speaker who would be willing to provide some editing work?

Response: We thank you for careful reading of our manuscript and kindly pointed out our grammatical errors. We used a manuscript editing service from a professional English editing company to improve the English writing of our manuscript. Since the editing resulted in many places of modifications, we did not mark all the English improvements in the revised manuscript. Instead, we highlighted the major revisions related to the responses to the editor's or the reviewers' comments and questions, with red font.

Some examples are listed below, but I stopped annotating after the first pages:

p3 line 50 affect->affecting
p3 line 53. Do you mean "blockbuster"?

Response: Yes, we do.

p4 line 66 "part of the effort has been to"
p4 line 80 "protease that interacted"

Response: All were corrected in the revised manuscript.

Minor points:

- For the overlap with clinical mutations, the authors should provide the following information: exactly for which part of the protein were predictions made? And were all clinical mutations in the entire protein included, or only those mutations which occur inside the region for which predictions were made?

Response: We used our method to predict drug-resistant mutations based on molecular docking. Therefore, only the residues in the binding region that directly affect the binding were predicted. We have stated and discussed this point in the revised manuscript (line 133-141; line 444-448).

- Figure 4; the axis labels are incorrect. The graph does not depict "IC50", but the ratio between the IC50 and that of the corresponding wild-type ABL; please make sure this is indicated on the axis label

Response: Thanks to this comment. This has been revised. The figure is now Figure 6.

- I think it would be good to include a summary of supplementary figures 8-11 as a main figure, it is an important result - perhaps as a bar graph of IC50s?

Response: We summarized these in bar graphs of K_D and IC_{50} respectively. Please refer to the new Figure 4 and Figure 5 in the revised manuscript.

- Supplementary figures 6 and 7 compare measurements for different mutations; but the scale of the axes is (slightly) different between the graphs. This makes it difficult to directly compare the data. Can the authors make all axes exactly the same scale?

Response: For different MST assay, the protein concentration and the fluorescence intensity used for the optimal response are different and therefore the responses cannot be directly compared to each other. The response is very sensitive (to 0.001 level). Though it seems that the scale are similar between 0.89 to 0.94, they actually differ a lot. If we make all axes exactly equal, some cure lines will be scaled to only occupy a small part of the figure. So we keep use of the current figure form.

- The same goes for supplementary figures 8-11.

Response: We keep the current figures for the same reasons mentioned above.

- The computation time is mentioned as a limiting factor. Can the authors make a brief mention somewhere of the typical computation time involved (e.g. number of cores used and hours)?

Response: For the simulations, we used 50 CPUs (Xeon E5 v2. Core code: Ivy Bridge EP) and each simulation took about 80 ~ 90 hours. This was added in the revised manuscript (line 178-180).

- At the end of page 15, the authors propose a strategy for finding new drug resistant mutations. Based on Figure 4; it seems to me that step #2 is not necessary, as only the IC₅₀ can already distinguish p-c from p-nc mutations. Is that indeed correct, or do I miss something? It is of course always good to check the kcat/km, and fig 4 provides useful information for the manuscript, but the kcat/km does not seem to add information for classifying observed vs non observed mutations.

Response: Our algorithm tried to simulate possible mutations that can alter the binding affinity of a drug, with the simplified hypothesis that the simulated mutations will not destabilise the protein structure and disable the protein function, for which we know are not always true. So we think step #2 is a required complement for our current computational method. Of course, this cannot be concluded from Figure 4 (now figure 6). Yes, you are right only the IC₅₀ can already distinguish p-c from p-nc mutations in figure 6. But kcat/km also provides information for classifying observed vs non-observed mutations, as we have stated in the revised manuscript (line 284-290): “Among the eight mutants tested, four of them had higher catalytic efficacy by more than six-fold compared to wt, which may not be tolerated by cells.”

- Supplementary Figure 1 mentions that the results were of statistical convergence. How was this tested?

Response: After repeated computation, we found that the top ranked mutations converged to some specific amino acid residues. The raw data can be found at <https://github.com/pkuljx/EVER/> (This URL was added in the Data Availability section). Our statistis was done on these data.

References:

- [1] Hou T, Zhang W, Wang J, Wang W. Predicting drug resistance of the HIV-1 protease using molecular interaction energy components. *Proteins*, 2009
- [2] Kevin Hauser, Christopher Negron, Steven K. Albanese, Soumya Ray, Thomas Steinbrecher, Robert Abel, John D. Chodera & Lingle Wang. Predicting resistance of clinical ABL mutations to targeted kinase inhibitors using alchemical free-energy calculations. *Communications Biology*, 2018.

Reviewer #3 (Remarks to the Author):

The authors demonstrate an algorithm to identify single point mutations of the ABL kinase domain that can predict drug resistance. Such an algorithm could contribute to drug design and is therefore worthy of consideration by researchers involved in identifying drugs to combat resistance to BCR-ABL inhibitors and potentially to drugs designed to circumvent resistance to other protein kinases. Specific points (not in any particular order):

- 1) Abstract. The claim that EVER might be of value to predict efficacy outside of a drugs safety window cannot be supported, since such a prediction would need prior information about target selectivity, preclinical animal testing and phase 1 clinical data.

Response: EVER was designed to predicted future drug-resitant mutations of marketed drugs. For marketed drugs, the drug safety windows are known. So we claimed that “Our suggested strategy for the prediction of drug-resistance mutations includes the computational prediction and *in vitro* selection of mutants with increased IC₅₀ values beyond the drug safety window.”

- 2) The Introduction should be redrafted to be more concise and focus more on BCR-ABL inhibitors to treat CML (e.g. HIV is not relevant to the current study). The need for target selectivity of drugs should be mentioned.

Response: Thanks to this suggestive comments. We added more introduction to BCR-ABL inhibitors to treat CML in the the revised manuscript (line 106-115). As our work is a combination of computation and *in vitro* experimental testing, the computaional methodology is very important. In order to do a comprehensive comparsion to current computaional methods, we did not remove the HIV part from the introduction section because most of the previous computaional work used HIV proteins as targets. The target selectivity problem of drugs is mentioned in the Discussion section at line 452-453. For example, the main target of axitinib is not BCR-ABL, so we did not do prediction for it.

- 3) The landmark study of imatinib resistance by Daley should be cited and discussed (Cell 2003;112:831).

Response: Thanks to this comment. This important work was cited and discussed. (line 59-61 and line 101-102)

- 4) The structural biology of ABL (active & inactive conformations) should be discussed, together with their relevance and extrapolation to BCR-ABL.

Response: The structural biology of ABL (active & inactive conformations) , together with their relevance and extrapolation to BCR-ABL, were discussed in the revised manuscript. (line 106-115)

- 5) The strategy to combat mutations by non ATP-competitive approaches (e.g. J Med Chem 2018;61:8120) or by targeting signalling nodes should be mentioned.

Response: The strategy to combat mutations by non ATP-competitive approaches (J Med Chem 2018;61:8120) and by targeting downstream signalling nodes (Bioorg Med Chem Lett 2015; 25: 4047-4056) were mentioned and cited in the discussion. (line 444)

- 6) The Results section includes much discussion (e.g. lines 193-213)which should be moved to the Discussion section. The assumption on line 127 is a major limitation, since mutations or detected outside of the ATP-binding site, as the situation with ponatinib (line 292), and these warrant discussion.

Response: Thanks for the suggestion. The content of lines 193-213 was moved to the discussion section (line 419-423 in the revised manuscript). Our method predicts drug-resistant mutations based on molecular docking. Therefore, only the residues in the binding region that directly affect the binding were predicted. We have stated and discussed this point in the revised manuscript. (line 444-448)

- 7) The comment on line 234 that mutations destabilise the protein is incorrect, since a destabilised protein will not drive disease. Mutants must be able to bind ATP and be functional in phosphorylating their substrates: the mutations destabilise particular protein conformation and this need to be discussed in detail (See Mini Rev Med Chem 2004;4:285).

Response: Our method did not consider disease-level issues, during computaional simulation. The algorithm tried to simulate any possible mutations that can alter the binding affinity of a drug, with

the simplified hypothesis that the simulated mutations will not destabilise the protein structure, for which we know it is not always true. We think some of the mutations our method supposed will not really happen by the reasons of protein destabilisation, and maybe some other reasons. That can explain why some mutations are predicted, but not found in the clinic.

- 8) Decreasing drug binding/potency can promote resistance, as shown nicely by Duyster et al (e.g. Fig 1 of Blood 2006;108:1328); the sentence (line 339) needs to be redrafted.

Response: Yes, you are right that decreasing drug binding/potency can promote resistance. That is also our point and is the basis of our simulation. We stated that “We also found that simply decreasing drug binding strength is not enough to produce drug resistance, as the enzyme activity change also needs to be considered.” The sentence is not to deny this point. The meaning of the sentence is that in most cases decreasing drug binding/potency can promote resistance, but in a few cases the enzyme activity change also needs to be considered. We re-write the sentence to “We have also found that simply decreasing drug binding strength is not enough to produce drug resistance, as the enzyme activity change also needs to be considered.” in the revised manuscript (line 432-434).

- 9) On line 337 the authors write that dasatinib can bind DFG-in & DFG-out ABL, but this is based on speculation and has been shown not to be the case experimentally (J Biol Chem 2008;283:18292) and this needs to have more discussion.

Response: Thanks for this comment. We have also reviewed related literatures and found NMR and MD-based evidences demonstrating that there is essentially no pattern of Dasatinib binding to the inactive conformation. (Mol Oncol. 2013 Oct;7(5):968-75) . We added discussion on this in the revised manuscript (line 106-115).

- 10) In the case of ponatinib drug-resistant compound mutations might be mentioned (see Blood 2016;127:703), where the authors could discuss the potential of EVER to predict these.

Response: Resistant mutations to Ponatinib are usually multi-point mutations. Only I315M is a prominent single-point resistant mutation. As our method was designed to predict single point mutations (for multi-point mutation prediction it will be not so reliable), we did not compare the predicted results to the clinically found mutations for ponatinib.

Some specific recommendations to text:

Line 22: ...clinically resistant...

Line 23: ...develop drugs to combat resistance....

Line 33: ...although...

Line 47: ...the 2050s. Developing drugs to combat resistance in cancer, such...

Line 51: In CML point mutations are the most common mechanism of resistance

Line 53: ...including important drugs such as...The authors should consistently use International Nonproprietary Names throughout, and these should not be capitalised.

Line 57: ...a solution...

Line 58: Mutations resistant to imatinib were predicted in advance by Corbin et al. Blood 2000;96:470a (Abstract#2025) & J Biol Chem 2002;277:32214.

Response: We discussed some experimental methods for drug-resistant variants prediction in line 59-65. J Biol Chem 2002;277:32214. was cited. But we cannot find the paper Blood 2000;96:470a.

Line 61: ...hindrance...

Line 64: Classify as intrinsic and extrinsic mechanisms.

Line 68: PCR does not generate mutations!

Response: We have revised the statement.

Line 102: ..protein that is not autoregulated and is always....

Line 105: ..inhibits the kinase activity of ABL.

Line 106: The authors should note that this was using mouse ABL. Do the authors use mouse or human ABL for their study

Response: We have noted this in the revised manuscript (line 100). We used the human ABL for our study. The NCBI reference sequence of human ABL is NM_005157.6 (line 511).

Line 107: ..clones harbour point...

Line 255: The steady-state plasma level of 400 mg BID nilotinib is not as written and this should be changed (see Eur J Clin Pharmacol 2012;68:723).

Response: The reference you mentioned is about the daily dose of nilotinib, which is somewhat different from the blood concentration written in our manuscript.

Line 281: ..for treating refractory CML...

Response: We thank you for your careful reading of our manuscript and kindly pointed out these above errors. We have corrected all of these in the revised manuscript. We also used a manuscript editing service from a professional English editing company to improve the English writing of our manuscript. Since the editing resulted in many places of modifications, we did not mark all the English improvement in the revised manuscript. Instead, we highlighted the revisions related to the responses to the editor's or the reviewers' comments and questions, with red font.

REVIEWERS' COMMENTS:

Reviewer #1 (Remarks to the Author):

none

Reviewer #2 (Remarks to the Author):

In my opinion the authors have adequately addressed the concerns. A better summary of the existing work is now included and the authors more clearly indicated the novelty of the present work and the reasoning behind their approach. The additional predictions and analysis for EGFR-gefitinib illustrate the extent of generalization of the method, and by keeping the scoring system and parameters exactly the same the authors addressed the potential issue of overtraining.